# Changes in Sleep Quality, Sleep Duration, and Sickness Absence: A Longitudinal Study with Repeated Measures

**DOI:** 10.3390/healthcare12141393

**Published:** 2024-07-11

**Authors:** Torbjörn Åkerstedt, Julia Eriksson, Sara Freyland, Linnea Widman, Linda L. Magnusson Hanson, Anna Miley-Åkerstedt

**Affiliations:** 1Department of Clinical Neuroscience, Karolinska Institutet, 171 77 Stockholm, Sweden; anna.miley.akerstedt@ki.se; 2Stress Research Institute, Department of Psychology, Stockholm University, 114 19 Stockholm, Sweden; linda.hanson@su.se; 3Department of Environmental Medicine, Karolinska Institutet, 171 77 Stockholm, Sweden; julia.eriksson@ki.se (J.E.); sara.freyland@ki.se (S.F.); 4Department of Medicine, Huddinge, Karolinska Institutet, 171 77 Stockholm, Sweden; linnea.widman@ki.se; 5Women’s Health and Allied Health Professionals Theme, Karolinska University Hospital, 141 86 Stockholm, Sweden

**Keywords:** Psychosocial, work, days off

## Abstract

Background: Sickness absence has been linked to short and long, as well as poor, sleep in a few studies. Such studies have started from a baseline measurement and followed up on subsequent sickness absence. In the present study, however, we focused on the *change* in biennial reports of sickness absence and sleep measures (using work-related variables as possible modifiers). We also searched for an interaction between predictors and gender since women report more sleep problems. Methods: A total of 5377 individuals (random sample from the Swedish working population) participated across five biennial points of measurement. Data were analyzed using mixed-model logistic regression. Results: The multivariable analysis of variation across the five time points showed that the significant sleep-related predictors of sickness absence (at least one occurrence during the preceding year) were sleep duration during days off (OR = 1.16, 95% Cl = 1.08;1.24) and sleep problems (OR = 1.42, 95% CI = 1.33;1.51). These also remained significant after the addition of psychosocial work factors. Sensitivity analyses indicated that a 9 h sleep duration during days off may represent a critical level in terms of increased sickness absence and that late rising contributed to the association between sickness absence and long sleep duration during days off. Women reported a higher sickness absence than men (OR = 2.16, 95% CI = 1.74;2.68) and had a higher probability of sickness absence for long sleep during days off and during the workweek than men. Conclusions: It was concluded that increases in sleep problems and sleep duration during days off are longitudinally associated with changes in sickness absence and that women have a closer link between the two. This suggests that treatment for sleep problems may reduce the risk of sickness absence.

## 1. Introduction

Poor sleep has been linked to a number of diseases [1], but less is known about its link to sickness absence. However, there is some support for a prospective link between sickness absence and sleep complaints [2,3,4]. Essentially, the risk of absence increases with the number or severity of sleep complaints. In addition, sleep duration (both long and short) has been linked to increased sickness absence in at least one study [5]. Interestingly, a number of diseases seem to be associated with both short and long sleep [6]. Both of the latter observations suggest a U-shaped association between sleep duration and sickness absence.

An important point regarding reported sleep duration is that all of the studies on, for example, sleep duration and mortality, ask for “usual” or “habitual” (or some similar expression) sleep duration as judged by the studies used in meta-analyses [7,8]. This suggests that reports of sleep duration are likely based on sleep during the workweek since that constitutes 5/7 of the week and would likely be the reference for most respondents. Since workweek sleep duration is likely to be truncated by work obligations, sleep duration during days off may yield additional information. However, we lack data on sleep during days off and sickness absence.

In general, one would expect increased sleep duration during days off to be associated with reduced sickness absence because of its apparent role as “catch-up” sleep [9]. In contrast, it has been shown that long (>8 h) sleep duration on both weekdays and days off (during baseline measurement) is associated with increased mortality, while short weekday plus long weekend sleeping is not [10]. The latter suggests that long sleep during days off may not solely be a positive factor, and it seems of interest to study the association between changes in sickness absence and changes in sleep duration separately for sleep during the workweek and during days off.

One should also be aware that work-related factors like job demands [11,12], a lack of job control [13], and physical workload [14,15] have been linked to sickness absence, but they have also been linked to sleep [16]. It, therefore, seems to be an interesting possibility to explore whether changes in work-related factors might moderate a putative link between changes in sleep and sickness absence.

Finally, gender may be of interest in the present context since women usually have a higher level of sickness absence [17], as well as sleep complaints [18]. Women also have a higher mortality associated with very short sleep (<5 h), and rate such sleep as having a lower quality, as compared to men [19]. The higher sickness absence in women is expected to be confirmed in the present study, but it might be the case that women are more sensitive to changes in sleep quality or sleep duration than men. Nothing is, however, known about this.

As stated above, there is evidence of a link between sleep and sickness absence. However, all prior work has measured exposure (sleep) at one point and compared sickness absence in individuals with poor and good sleep quality or individuals with short (typically < 6 h) and long (typically 9–12 h) sleep duration with a putative “normal” sleep duration (usually 7 h or 8 h). While valuable, this approach disregards observations that sleep varies across time [20]. It is also sensitive to confounding since exposure is based on between-group data. Here, we propose another approach, that is, to study whether a *change* in sickness absence is linked to a *change* in sleep quality or sleep duration. This approach using change includes both increases and decreases in the studied variables, whereas the traditional study evaluates the prediction of an increase in sickness absence. This approach also reduces confounding, even if it does not eliminate it entirely. We believe the approach of analyzing change may increase the understanding of the links between sleep and sickness absence.

The purpose of the present study was to examine changes in sickness absence (none vs. one or more days) across five biennial points of measurement and its association with changes in sleep quality ratings and sleep duration (for weekdays as well as during days off), as well as the possible influence of work demands, work control, and physical workload on the associations between sleep and sickness absence. We also investigated whether gender would interact with any of the predictors. None of the questions mentioned have been addressed before.

## 2. Methods

### 2.1. Design and Participants

This study was based on the Swedish Longitudinal Occupational Survey of Health, SLOSH. It is a longitudinal cohort study of individuals from the entire country, stratified by county, gender, and citizenship [21]. This survey has its origins in the Swedish Work Environment Survey (SWES, www.scb.se), which, in turn, is based on nationally representative samples of the working population in the age range of 16–64 years. The data collection (postal questionnaire) took place every two years in early/late spring. The participants were asked to respond to all questions, and reminders were sent out to those who did not respond. In 2006, the SLOSH cohort included 9214 individuals. The procedure was repeated every two years, in 2008, 2010, 2012, and 2014. Over 50% of those eligible at follow-up occasions in 2006–2014 responded to questionnaires. After the exclusion of individuals not working, those working less than 30%, and those without any information on sickness absence, 5377 individuals remained. No other exclusions were made. Note that the five biennial measurements were not used for analysis across successive years (see the section on statistical analysis). The sample at T1 contained 45.3% males, 57.5% who had at least one day of sick leave during the preceding year, and 40.2% blue-collar workers. The mean age ± SD was 45.3 ± 9.1 years. The study was approved by the Regional Ethics Review Board in Stockholm.

### 2.2. Variables

Information regarding gender and age was obtained from administrative registers. Other information was obtained through the SLOSH questionnaire. The occupational group (blue-collar worker, white-collar worker, and manager) was derived from self-reports at T2. Sickness absence was assessed with the questions: “How many times have you been sickness absent ≤ 1 one week during the preceding 12 months?” “How many times have you been sickness absent > 1 week during the preceding 12 months?”. We combined these two variables into a variable that represented any sickness absence vs. none (1/0).

Four items representing sleep problems were selected from the Karolinska Sleep Questionnaire (KSQ) (Cronbach’s alpha > 0.70) [22,23,24,25]. The scale differentiates patients with insomnia from healthy individuals [23], and correlates with perceived stress, anxiety, depression, and burnout (r > 0.40) [25]. The items included are difficulties falling asleep, restless sleep, repeated awakenings, and premature awakening. The responses range from “never” to “most days of the week” (1–6). Cronbach’s alpha was 0.85 at time T3, and the correlation between time points was r = 0.71. The questionnaire also includes separate items on sleep duration during workdays and during days off. The response was given in hours and minutes. Sleep duration during weekdays and days off was obtained from reports of bedtime (“switching off the lamp”) and time of rising (given in hours and minutes). It should be pointed out that both sleep duration variables refer to sleep before a day of work or before a day off, respectively, whereas the day a sleep was started could be either a workday or a day off.

Work demands and control at work were measured using the Demand–Control–Support Questionnaire [26], based on work by Karasek et al. [27]. This scale has been extensively psychometrically investigated [28,29] and used to predict health outcomes of psychosocial work factors [30,31]. The five items used for work demands were focused on whether the respondent had to work very hard, very fast, had enough time for tasks, or was exposed to conflicting demands [28]. The response alternatives ranged from 1 = ”hardly ever/never”, to 4 = “Yes, often” (scoring is reversed from the original 4-1, except for the item about having enough time). Cronbach’s alpha was 0.74 at T3. The items used for control at work were focused on whether work is repetitive, allows freedom to decide how to work, requires skill and creativity, or involves learning new things [28]. The response alternatives ranged from 1: “Yes, often”, to 4: “Hardly ever/never”. Cronbach’s alpha was 0.56 at T3. We named this variable “lack of control”.

Physical workload was measured as an index of three questions: “Is your work such that you have to use bent, twisted or otherwise unsuitable positions?” “Do you have to lift at least 15 kilos several times a day?” “Does your work sometimes involve heavy physical labor, that is, do you physically exert yourself more than when walking and standing and moving around in a normal way?”. Response alternatives ranged from 1: = “no, not at all” to 6 = “most of the time”. Cronbach’s alpha was 0.90 at T3.

### 2.3. Statistical Analysis

To assess the connection between sleep and sick leave over time (waves T1–T5), mixed-effects hierarchical logistic regression models were used [32]. The mixed-effects approach has the advantage that it allows for simultaneous analysis of longitudinal and between-groups components. In addition, it is not sensitive to loss of data, so long as it is “missing at random”. Thus, no individuals were excluded because of missing data. It should be emphasized that the independent variable here was sleep (or work factors) and the dependent variable was sickness absence. This means that “wave ” or “year” was not a predictor. Instead, each wave contributed, for each individual, one pair of values, for example, one value for sleep quality and one for sickness absence. The five sets of pairs from the five waves were then used to form the regression. The regression models included a subject-specific random intercept and a random slope to account for potential dependence among repeated values of sick leave over time. The analyses were adjusted for age and occupational group. Gender was treated as a fixed factor and included in all analyses. The full range of the response scale was used in all analyses, except for sickness absence (see above). The results are presented in four models. Model 1 contains the results for the univariate analysis (only adjusted for gender (but not for gender as a predictor), age, and occupational group). In model 2, the sleep variables were entered. In model 3, the work-related predictors were entered. Multiplicative two-way interactions for each predictor and gender were tested in the completely unadjusted regression models (all interactions were tested). The analyses were computed in Stata 17 (StataCorp. 2021. Stata Statistical Software: Release 17. College Station, TX, USA: StataCorp LLC.)

## 3. Results

Table 1 describes the mean and standard deviation (or percent) for background factors and absence, as well as for predictors averaged across time for each individual. Notably, at least one day of sick leave was reported by 57.9% (higher in women). In Appendix A, the mean longitudinal standard deviation (SD) is given for each variable to reflect variation over time. For sleep duration during the work week, the SD was 0.51 h. Corresponding values for sleep duration during days off was SD = 0.59 h and for poor sleep quality was SD = 0.56 units.

Table 2 shows the results from the hierarchical mixed-effects logistic regression. In Model 1, all ORs were significant. Sickness absence increased with most of the significant changes in the predictors. Model 2 shows that increased sleep problems and sleep duration during days off were significantly associated with increased sickness absence (which was true for all models. In Model 3 (work variables), the significant variables became lack of work control and physical workload (both increased with increased sickness absence), while the variables significant in Model 2 remained significant. We also examined absence > 1 week and absence 1–7 days as separate dependent variables, but the results differed only marginally from those in Table 2.

The multiplicative interaction term between gender and predictors (with sickness absence as the dependent variable) was significant for gender and sleep duration during days off, with OR = 1.18 (95% CI = 1.03;1.35, *p* < 0.05) (Figure 1). Females showed a sharper increase than males in sickness absence with increasing sleep duration during days off (both significant, though). For the workweek, the interaction term was OR = 1.23 (95% CI = 1.04;1.44, *p* < 0.05). The separate regression for each gender was not significant (Figure 1). Also, the interaction between gender and lack of work control showed a significant interaction, with OR = 0.77 (95% CI = 0.59;0.97, *p* < 0.05). Thus, men showed a lower probability of sickness absence than women with an increasing lack of control.

The percentage of individuals with long sleep (≥9 h) was 8.7% for the workweek and 44.0% for days off. In order to ascertain if there was a critical sleep duration for increased risk of sickness absence, we reanalyzed the association between sleep duration and sickness absence excluding sleep durations ≥13 h, ≥12 h, and ≥7 h, and found that the association was significant only when sleep duration exceeded 9 h during days off.

Since the measures of sleep duration were derived from bedtime and time of rising, we conducted an explorative analysis to determine the association of changes in only the latter variables with changes in sickness absence (adjusted for gender, age, and occupation). This yielded OR = 1.19 (CI = 1.12;1.27, *p* < 0.001) for the time of rising on days off and OR = 0.91 (CI = 0.85;0.98, *p* < 0.05)) for bedtime during days off. For workweek sleep durations, the result was not significant (OR = 98 (CI = 0.92;1.04, ns) and OR = 0.95 (CI = 0.89;1.01, ns), respectively). Thus, sickness absence increased with later rising on days off and with earlier bedtimes during days off.

As a sensitivity analysis, we repeated the main analysis with only those subjects that had complete data for all five points of measurement (N = 1500). This did not change the results, except for the loss of significance for the two previously significant interaction effects.

## 4. Discussion

Sickness absence increased when sleep problems and sleep duration during days off increased. In addition, an increased lack of control at work was associated with increased sickness absence. Women had a significantly higher sickness absence across all models and the association was stronger for women.

As indicated in the introduction, there are no prior studies of change in sleep problems and their relation to change in sickness absence, but we may gain some information from studies that find an association between interindividual differences in baseline sleep and sickness absence during the following year(s) [2,3,4], that is, studies in which poor sleepers are compared to good sleepers. In the present study, we found that a relatively modest (see Table 1) increase in sleep problems was associated with an increased probability of sickness absence. The results strengthen the notion of an association between changes in poor sleep and changes in sickness absence and suggest that an intraindividual increase in sleep problems may serve as a warning sign of increased sickness absence. We cannot draw conclusions on causation, however, since there are several possible interpretations. It might be that the effects of poor sleep lead to a need to report sickness in order to recover. There is also the possibility that the presence of a disease may cause both disturbed sleep and sickness absence.

Also, the significant association between increased sleep duration during days off and increased sickness absence lacks comparative studies. Resorting to other types of designs again, we find that Lallukka et al. [5]. showed a higher sickness absence in individuals with long sleep (9–12 h) than those with “normal” sleep, which would lend some support to the results of the present study [5]. As with the increase in sleep problems, we find here that a modest increase in sleep duration during days off is linked to an increase in the probability of sickness absence. As discussed above, no conclusions on causation can be drawn from the present results, but one interpretation may be that changes in long sleep may herald increased sickness absence because of latent disease causing both long sleep and sickness absence since sleep and disease are highly interrelated [33,34]. In the literature on long sleep and mortality, undiagnosed disease (inflammation) is often offered as a putative explanation of the link [7]. We also found that the change in sleep duration needed to reach the >9 h sleep duration segment to be significant and that 46% of the sleep episodes exceeded 9 h, permitting long sleep to have an impact in the analyses. Thus, this suggests that 9 h of sleep duration may constitute a critical value for the occurrence of sickness absence and possible other indicators of ill health. This, however, needs confirmation in other studies.

As with the previous two predictors, the results for changes in sickness absence and sleep duration during the workweek lack comparable studies. The lack of association seems relatively logical, however, considering the apparent truncation of sleep duration due to work, with very few people reporting sleep durations exceeding 9 h (8.7%), and a mean duration of 7.52 h, close to what may be optimal sleep duration in studies investigating habitual sleep duration and sickness absence [5,6]. It should be emphasized that the dataset did not include very short sleep durations (mean of 7.47 ± 0.80 h), which would prevent any conclusions on the effects of changes in the very short sleep range (≈<5 h).

The significant interaction of gender and sleep duration adds a further dimension to the discussion above since women’s sickness absence appears more closely linked to increases in sleep duration than that of men. We cannot explain the reason for this, but the observation fits a general pattern of higher prevalence of disturbed sleep and psychiatric problems in women, as suggested by Voderholzer et al. [35]. The higher overall risk of sickness absence in women was expected from previous work [17,36]. One may also consider the influence of social factors. Women tend to have a higher responsibility for family issues [34].

The explorative analysis of bedtimes and rise times suggests that the link to increased sickness absence during days off is particularly due to late rising and to a lesser extent to earlier bedtimes. The observation on the mean latest rising (from 7.53 ± 1.07 h to 8.57 ± 1.07 h) and increased sickness absence is reminiscent of the link between eveningness and impaired health [37]. This suggests that chronotype could be an interesting variable in future studies of sickness absence.

The observation that changes in work factors did not affect the association between changes in sleep and sickness absence was somewhat unexpected, considering the literature linking such factors to both sleep and sickness absence [11,12,13,14,15]. Apparently, the links between sleep and sickness absence were independent of the analyzed work-related factors. However, the significant association between change in sickness absence and increase in lack of work control and increase in physical workload appears to strengthen the link between work factors and sickness absence. The significant interaction between gender and lack of control suggests that women have a stronger increase in the probability of sickness absence than men. This may be due to the factors discussed earlier regarding sleep duration but goes beyond the purpose of the present study.

The present study has some limitations. Firstly, all data are based on self-report. Register data on sickness absence may have been preferable. However, official registers in Sweden and many other countries often only include longer bouts of absence (two weeks in Sweden). Thus, short-term absence is not recorded but is available through self-report. In addition, we have no information on other factors that may affect sickness absence, for example, responsibilities for family and relatives. Certainly, such factors may affect both sleep variables and sickness absence.

In the present study, only individuals at work were included, but the criterion for work was set at 30% of full-time. Thus, some individuals may have had more opportunities for sleep than others, which may have affected associations. Incidentally, the study by Lallukka [5] included all individuals who had had *any* work during the previous year. Another problem may be that we do not know what measurement interval is optimal in a prospective study of this type, while the present study used biennial measurements. More frequent measurements may be preferable. In a longitudinal study, there is also a certain amount of attrition because of loss of interest, time constraints, health issues, or deaths. Particularly, the two latter factors mean a selection of healthy individuals, which may lead to a selection bias. This would likely work against the hypothesis of a link between poor/short sleep and sickness absence. The present study is correlative, even if prospective. We, therefore, have no possibility of inferring causality. Finally, the present focus was on sleep as a predictor of sickness absence, beyond contributions of age, gender, occupation, psychosocial, and physical work variables. However, there is a possibility of confounding by other factors, such as lifestyle or responsibility for family or relatives. The inclusion of such variables may give a broader picture of factors associated with change in sickness absence. This was beyond the focus of the present paper.

## 5. Conclusions

In conclusion, the present study has shown that increased sleep problems and sleep duration during days off are associated with increased sickness absence across five biennial measurements, also while adjusting for other predictors of sickness absence like age, occupation, health, and work environment variables. Women had a closer link than men between increased sleep duration during days off and increased sickness absence. The findings suggest that increases in sleep problems and sleep duration during days off may serve as risk indicators of sleep loss, and treatment for sleep disorders may reduce the risk of sickness absence, and perhaps also chronotype.

## Figures and Tables

**Figure 1 healthcare-12-01393-f001:**
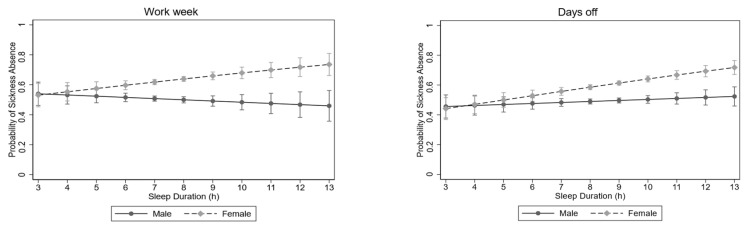
Association between changes in sleep duration and changes in sickness absence during the workweek (**left**) and during days off (**right**) for males and females. OR and 95% Confidence interval. Each individual is represented by several points that describe the sickness absence (or not) for different sleep durations, that is, a regression line for each individual. The regression line depicted in the graphs represents the combinations of these individual regression lines.

**Table 1 healthcare-12-01393-t001:** Background data. Mean ± SD of background variables, sick leave, and across measurements (based on the mean of each individual).

	Average across Time: Mean ± SD across Individuals N = 5377	Average across Time: Mean ± SD across Individuals Males N = 2437	Average across Time: Mean ± SD across Individuals Females N = 2940
Age	46.9 ± 11.1	47.2 ± 11.1	46.6 ± 11.26 ^a^
Blue collar	40.2%	47.1%	43.6% ^c^
Subjective health (1–5 high)	4.0 ± 0.81	3.98 ± 0.80	4.02 ± 0.81
Total sickleave (yes)	57.9%	50.1%	64.4% ^c^
Work demands (1–4 high)	2.63 ± 0.50	2.60 ± 0.50	2.65 ± 0.52 ^c^
Lack of work control (1–4 low)	1.93 ± 0.43	1.90 ± 0.44	1.95 ± 0.41 ^c^
Physical workload (1–6 high)	2.30 ± 1.43	2.47 ± 1.52	2.15 ± 1.34 ^c^
Poor sleep quality (1–6 poor)	2.55 ± 0.96	2.39 ± 0.91	2.58 ± 0.99 ^c^
Sleep duration: work (h)	7.47 ± 0.80	7.28 ± 0.81	7.63 ± 0.74 ^c^
Sleep duration: days off (h)	8.66 ± 0.93	8.46 ± 0.97	8.82 ± 0.87 ^c^
Bedtime work	22.69 ± 1.24	22.84 ± 1.23	22.57 ± 1.24 ^c^
Time of rising work	6.16 ± 1.38 h	6.14 ± 1.49	6.19 ± 1.27
Bedtime days off	23.42 ± 0.96	23.56 ± 1.00	23.30 ± 0.90 ^c^
Time of rising days off	0.07 ± 1.12	8.03 ± 1.19	8.11 ± 1.05 ^b^

For differences between males and females, ^a^ = *p* < 0.05, ^b^ = *p* < 0.01, ^c^ = *p* < 0.001.

**Table 2 healthcare-12-01393-t002:** Results from hierarchical mixed-effects logistic regression across T1 to T5, vs. sickness absence (>0 times vs. no sickness absence) during the last 12 months. Models 1–3 are adjusted for age, gender, and occupational group.

	Model 1 OR 95% CI/Constant Single Variables N = 5207–5377	Model 2 Sleep OR 95% CI N = 5145	Model 3 Work OR 95% CI N = 5048
Female (vs. male)	2.07 ^c^ 1.69;2.76/0.42	2.00 ^c^ 1.72;2.32	1.97 ^c^ 1.70;1.48
Sleep problems (1–6 high)	1.42 ^c^ 1.35;1.51/0.68	1.42 ^c^ 1.33;1.51	1.39 ^c^ 1.31;1.48
Sleep duration workweek (h)	1.14 ^b^ 1.05;1.24/0.63	0.99 0.90;1.08	1.02 0.93;1.11
Sleep duration days off (h)	1.19 ^c^ 1.11;1.27/0.38	1.16 ^c^ 1.08;1.24	1.15 ^c^ 1.07; 1.23
Work demands (1–4 high)	1.19 ^c^ 1.08;1.30/1.05		0.98 0.88;1.09
Lack of work control (1–4 low)	1.82 ^c^ 1.60;2.06/0.52		1.64 ^c^ 1.43;1.89
Physical workload (1–6 high)	1.17 ^c^ 1.12;1.22/1.16		1.15 ^c^ 1.09;1.21
Constant		0.19	0.07

^b^ = *p* < 0.01, ^c^ = *p* < 0.001.

## Data Availability

This work utilized data from SLOSH, which is part of the REWHARD consortium, supported by the Swedish Research Council (VR #2021-00154). Given restrictions from the ethical review board and considering that sensitive personal data are involved, it is not possible to make the data freely available. Access to the data may be provided to other researchers in line with Swedish law and after consultation with the Stockholm University legal department. Requests for data, stored at the Stress Research Institute, Department of Psychology, should be sent to data@slosh.se.

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
