# Peer review of "Changes in Sleep Quality, Sleep Duration, and Sickness Absence: A Longitudinal Study with Repeated Measures"

_healthcare, 2024, doi:10.3390/healthcare12141393_

Round 1

Reviewer 1 Report

Comments and Suggestions for Authors

Line 50, please check the wording: “that it is seems

Line 54: this aspect is new to me, very interesting; since we used actigraphy, we always include a weekend because of social jetlag. Perhaps you can be more precise if really all people include only the workweek? This might be an enormous shortcoming in previous studies, but I know that in most survey studies, measures like the MCTQ are used to show differences between workdays and free days. Perhaps you can elaborate a bit on this important aspect.

Line 100 bracket missing

Line 136: Statistical analysis: @editor: please refer here to other reviewers; I am not an expert on these models, but by reading the text they seem reasonable.

The interactions are a bit confusing: did you assess all interactions, only two-ways, or die dyou delete some interactions (non-significants); there were many ways on the way to Rome, and may correct. Please specifiy a bit more.

What does the OR mean in Table 1, perhaps I am not smart enough but it seem Mean and SD?

Line 267: fully agree, in some companies, it is not centrally documented if a person is absent for one day (in some countries) and it depends on the law and the internal regulations.

Line 285: it should be shortly mentioned also that chonotype is an important predictor of health outcomes, and as far as I can see, it could be easily calculated using Roenneberg’s concept. It would be even an important future study to see whether it is sleep duration, chronotype or sleep quality that has the strongest effect. Especially in line 288 where you again emphasize the sleep duration in days off, which is usually a clear indicator for catch-up sleep of late chronotypes.

Author Response

  1. Line 50, please check the wording: “that it is seems”.
    1. R: Done.
  2. Line 54: this aspect is new to me, very interesting; since we used actigraphy, we always include a weekend because of social jetlag. Perhaps you can be more precise if really all people include only the workweek? This might be an enormous shortcoming in previous studies, but I know that in most survey studies, measures like the MCTQ are used to show differences between workdays and free days. Perhaps you can elaborate a bit on this important aspect.
    1. R: You are right that in studies that focus on chronotype one obtains data, as you say. But in epidemiological studies on sleep and mortality or disease, this is not the case (there may be exceptions, but I can’t recall any). The case is the same with survey studies of sleep problems. We don’t have the possibility to make a major review of the issue, but my impression is, as we suggest in the text, that questions on sleep duration refer to “usual sleep duration”, “normal sleep duration”. It may also be based on “when do you normally go to bed /or turn out the light for sleep), and when do you normally wake up. This is definitely the case with studies on sleep duration and sickness absence.
  3. Line 100 bracket missing.
    1. R: Now added.
  4. Line 136: Statistical analysis: @editor: please refer here to other reviewers; I am not an expert on these models, but by reading the text they seem reasonable.
    1. R: Can’t really do anything here, of course, but the method has been “in vogue” for a number of years. But, to increase clarity we added the following to the section on “Statistical analyses”: The mixed effects approach has the advantage that it allows for a simultaneous analysis of longitudinal and between-groups components. In addition it is not sensitive to loss of data, as long as it is “missing at random”.
  5. The interactions are a bit confusing: did you assess all interactions, only two-ways, or die dyou delete some interactions (non-significants); there were many ways on the way to Rome, and may correct. Please specifiy a bit more.
    1. R:Thank you, that is probably needed. We should have been more informative here. We have now add text in the section on “Statistical analyses”, saying that we evaluated all two-way interactions, but we don’t present the non-significant results, it would mess up the text since there were many of them.
  6. What does the OR mean in Table 1, perhaps I am not smart enough but it seem Mean and SD?
    1. R: Aha, you are right, of course. It was cut and pasted from another table and not properly checked. Now removed
  7. Line 267: fully agree, in some companies, it is not centrally documented if a person is absent for one day (in some countries) and it depends on the law and the internal regulations.
    1. R: Yes, indeed.
  8. Line 285: it should be shortly mentioned also that chonotype is an important predictor of health outcomes, and as far as I can see, it could be easily calculated using Roenneberg’s concept. It would be even an important future study to see whether it is sleep duration, chronotype or sleep quality that has the strongest effect. Especially in line 288 where you again emphasize the sleep duration in days off, which is usually a clear indicator for catch-up sleep of late chronotypes.
    1. R: We did mention eveningness on line 257, and have now added a second sentence: This suggests that chronotype should be an interesting variable in future studies of sickness absence. We also added “perhaps also chronotype” to the last sentence (257) in the conclusion.

Reviewer 2

    1.  

Reviewer 2 Report

Comments and Suggestions for Authors

First of all, I think the work has the potential to be interesting to readers and a lot of effort has gone into it. However, I have some concerns and suggestions;

First of all, the data is presented in a somewhat scattered manner in the introduction section. You can navigate in the order specified in the title. Apart from this, sickness absence can also be defined. In this way, the reader can reach a clearer understanding.

Minimum data are presented in the method section. What were the inclusion and exclusion criteria? Are there more details about the demographic data of the participants? Evaluation of sleep duration based on declaration raises some concerns. Since individuals perceive their bedtime and wake-up time to be much wider, measurements made with devices such as Android smart watches can show that individuals sleep less than they declare. This may be because patients may wake up at night and stay awake for a certain period of time.

In the method section, was it asked how the study was conducted, how often and in what periods the measurements were made, and whether each measurement tool was required to be filled in each period? Did all participants complete the surveys at all times? Were there any losses? If so, what was done about these losses? Were they included or excluded from the analyses? Etc. More details should be given on issues such as.

Line 85 should be replaced with the remaining quote number.

In the results section, measurements can be given separately according to time in the tables. This will be more meaningful in evaluating the effects of seasonal differences on sleep and diseases.

I disagree with the suggestion in line 219. Taking more frequent measurements may not be enough to understand why. To understand this, different questions should be asked or a method such as mixed methods research may be preferred.

There may be different factors that can affect sleep and sickness absence. How were these brought under control? If individuals are leaving not because of their own illnesses but because of the illnesses of their relatives (parents of addicted children, etc.), what distinction is made in this regard?

What is the possibility that women's social absence is due to the fact that they are given different roles compared to men? This can be added to the discussion section.

Author Response

  1. First of all, I think the work has the potential to be interesting to readers and a lot of effort has gone into it. However, I have some concerns and suggestions;
  2. First of all, the data is presented in a somewhat scattered manner in the introduction You can navigate in the order specified in the title.
    1. R: We have tried to create a better order, but it is partly a matter of taste.
  3. Apart from this, sickness absence can also be defined. In this way, the reader can reach a clearer understanding.
    1. R: We have now added this to the purpose.
  4. Minimum data are presented in the method section. What were the inclusion and exclusion criteria?
    1. R: In the first section of Methods, we stated that those working full or part-time (>30%) were included. Now we have added that no other exclusions were made.
  5. Are there more details about the demographic data of the participants?
    1. R: We don’t have much more demographic data, but now we have added subjective health.
  6. Evaluation of sleep duration based on declaration raises some concerns. Since individuals perceive their bedtime and wake-up time to be much wider, measurements made with devices such as Android smart watches can show that individuals sleep less than they declare. This may be because patients may wake up at night and stay awake for a certain period of time.
    1. R: Yes, we are aware of the weaknesses of self-reported sleep, and this is included in the discussion of limitations.
  7. In the method section, was it asked how the study was conducted, how often and in what periods the measurements were made, and whether each measurement tool was required to be filled in each period? Did all participants complete the surveys at all times?
    1. R: We have now added (line 85-86) that “The data collection took place each year in early/late spring and the participants were asked to respond to all questions”
  8. Were there any losses? If so, what was done about these losses? Were they included or excluded from the analyses? Etc.
    1. R:The approach to losses is discussed under “Statistical analyses”. No individuals are excluded because of missing data. As now stated in the description of the mixed model regression, the method is not sensitive to missing data (which is one of its major strengths), as long as the data are missing at random. However, we have now added a repetition of the analysis in table 2 using only those participants who hade complete data on all 5 points of measurement (N=1500).
  9. More details should be given on issues such as.
    1. R: Something is missing here
  10. Line 85 should be replaced with the remaining quote number.
    1. R: Don’t really understand what the “the remaining quote number refers to”. If you mean the reference to the site, I believe it should be in the text, but I am not absolutely certain. Maybe the technical editor may decide.
  11. In the results section, measurements can be given separately according to time in the tables. This will be more meaningful in evaluating the effects of seasonal differences on sleep and diseases.
    1. R: All repeated measures were obtained at the same time of year (as we now have added to the methods section), so there is actually no temporal data to display that can be used for studying seasonal effects. We can certainly add mean±se for each year, but that would be irrelevant to the purpose. What we are investigating is how each individual moves up or down a predictor across years and how that relates to a corresponding movement of the outcome (sickness absence) – this movement between years is not connected the specific calendar years, but will constitute a different temporal pattern for different individuals.
  12. I disagree with the suggestion in line 219. Taking more frequent measurements may not be enough to understand why. To understand this, different questions should be asked or a method such as mixed methods research may be preferred.
    1. R: Yes, you are probably right, sentence removed.
  13. There may be different factors that can affect sleep and sickness absence. How were these brought under control? If individuals are leaving not because of their own illnesses but because of the illnesses of their relatives (parents of addicted children, etc.), what distinction is made in this regard?
    1. R: You are right, but we did not ask for such information, so we cannot really provide more knowledge. We have, however, added a sentence to limitations to that effect: In addition, we have no information on other factors that may affect sickness absence, for example, responsibilities for family and relatives. Certainly, such factors may affect both sleep variables and sickness absence. We have now added some sentences to limitations.
  14. What is the possibility that women's social absence is due to the fact that they are given different roles compared to men? This can be added to the discussion section.
    1. R: Yes, is is a highly relevant observation, and we have now added a sentence in the discussion: “One may also consider the influence of social factors. Women tend have a higher responsibility for family issues.34.

Round 2

Reviewer 2 Report

Comments and Suggestions for Authors

The authors made the necessary edits to the article. Thank you, good job.

In the 1st revision, "In the Method section, were questions asked about how the study was conducted, how often and in what periods the measurements were made, whether each measurement tool had to be completed in each period? Did all participants fill out the questionnaires at all times?" I think the answer to the question needs to be developed a little more. The authors have made additions about how the study was conducted, but I think they should write a more detailed explanation on this subject. Research articles should be written in a reproducible manner. Therefore, authors should write details about the research process. Although it is clear how the article works as a whole, a reader who comes to the findings after reading the introduction and method gets the feeling that something different has been done in the article. For this reason, information about the process should be detailed in the method section.

Throughout the article, citations are shown in upper numeral format. For this reason, the reference to "(Magnusson Hanson et al., 2018)" in line 85 in the previous revision and line 95 in the new revision should be displayed numerically.

Thank you, Best Regards

Author Response

Thank you of your comments.

Firstly, we have turned the Magnusson Hansson citation into its proper form.

Then for the more complex issue. The reviewer is finds information on the procedure lacking. We are not sure that we fully understand what is lacking, but could it be that we haven’t explained clearly engough that we use a mixed model regression analysis. That is, we don’t analyze the change across the successive 5 time points (“years” or “waves”), even if we certainly collect data that way. It may be that this formulation gives the expectation that “years” (or waves) is the main predictor in the analyses. It is not. We now have added text in the first section of methods and in the last (statistics), explaining that each year gives us two values for each individual (e.g. one for sleep and one for sickness absence) and that we compute a regression across the pairs of observations (sleep vs sickness absence). Successive years (or waves ) do not enter into the analysis.

We do hope this clarifies the procedure. If not, we will be happy to respond again.

Sincerely